# Seasonal Variations of Rosmarinic Acid and Its Glucoside and Expression of Genes Related to Their Biosynthesis in Two Medicinal and Aromatic Species of *Salvia* subg. *Perovskia*

**DOI:** 10.3390/biology10060458

**Published:** 2021-05-22

**Authors:** Marta Stafiniak, Sylwester Ślusarczyk, Bartosz Pencakowski, Adam Matkowski, Mehdi Rahimmalek, Monika Bielecka

**Affiliations:** 1Department of Pharmaceutical Biology and Biotechnology, Wroclaw Medical University, Borowska 211, 50-556 Wroclaw, Poland; marta.stafiniak@umed.wroc.pl (M.S.); sylwester.slusarczyk@umed.wroc.pl (S.Ś.); bartosz.pencakowski@umed.wroc.pl (B.P.); 2Department of Horticulture, College of Agriculture, Isfahan University of Technology, Isfahan 841583111, Iran; mrahimmalek@cc.iut.ac.ir; 3Botanical Garden of Medicinal Plants, Wroclaw Medical University, Jana Kochanowskiego 14, 51-601 Wroclaw, Poland

**Keywords:** phenylpropanoids, phenolic compounds, rosmarinic acid, salviaflaside, salvianolic acid L, Russian sage, gene expression analysis, RAS1, RAS2, Cyp98A14

## Abstract

**Simple Summary:**

Here, we studied two closely related medicinal and aromatic plants from Asia, called Russian sage or from their previously used Latin name–Perovskia. These plants contain various specialized metabolites called phenylpropanoids that contribute to their medicinal uses. In our experiments, several different specialized phytochemicals were traced down in the roots and leaves with the major metabolite called rosmarinic acid, known for health beneficial properties. In order to check if the composition of these plants is regulated by specific genes encoding proteins that assemble these phytochemicals, we analyzed their expression during the growth season (spring, summer and fall). Despite being the closest kin, the two species of Russian sage displayed different seasonal changes in the composition of bioactive metabolites and the activity of genes responsible for their production. The genes’ activity was correlated with rosmarinic acid content in the roots but not in the green parts of the plants. Two genes pointed out were linked to the regulation of rosmarinic acid biosynthesis, called RAS (for Rosmarinic Acid-Synthase) and a newly reported version of an oxidizing enzyme called Cyp98A14. These discoveries broaden our understanding of relationships between the genes’ activity and production of bioactive constituents in herbs such as the two studied species of Russian sages.

**Abstract:**

*Salvia abrotanoides* Kar. and *Salvia yangii* B.T. Drew are medicinal and aromatic plants belonging to the subgenus *Perovskia* and used as herbal medicines in Asia. Derivatives of caffeic acid, mainly rosmarinic acid (RA), are the major phenolic compounds identified in these plants. Understanding the factors and molecular mechanisms regulating the accumulation of pharmacologically and ecologically relevant phenolic metabolites is essential for future biotechnological and medical applications. Up to date, no studies of phenylpropanoid biosynthetic pathway at the transcriptional level has been performed in the *Perovskia* subgenus. Using a combined qRT-PCR transcriptional activity analysis with LC-MS based metabolic profiling of roots and leaves at the beginning, in the middle and at the end of vegetation season, we have identified the following gene candidates with properties correlating to phenolic acid biosynthesis in *S. abrotanoides* and *S. yangii*: *PAL*, *C4H*, *4CL*, *TAT*, *HPPR*, *RAS1*, *RAS2* and *Cyp98A14*. A comparison of phenolic acid profiles with gene transcript levels revealed the transcriptional regulation of RA biosynthesis in the roots but not the leaves of the studied species. Additionally, *RAS1* and *Cyp98A14* were identified as rate-limiting steps regulating phenylpropanoid biosynthesis on a transcription level. In the future, this will facilitate the gene-based metabolic enhancement of phenolic compounds production in these promising medicinal herbs.

## 1. Introduction

Phenolic compounds, possessing various pharmacological activities such as anticancer, antioxidant, antibacterial, antiinflammatory, cardioprotective and neuroprotective activities, are aromatic secondary metabolites widespread in the plant kingdom [1,2]. Phenolic compounds also exhibit nutraceutical value, mainly due to their antioxidant capacity [3,4]. Although phenolic compounds are extensively reported in different plant taxa, Lamiaceae has been recognized as a potential source of these health-promoting chemicals [5,6]. Numerous studies have demonstrated various bioactivities and pharmacological potential of phenolic compounds from *Salvia miltiorrhiza* Bunge, commonly used in traditional Chinese medicine [7,8]. There are more than 20 phenolic acids in *S. miltiorrhiza*, including rosmarinic acid (RA), which is a typical product of the Nepetoideae subfamily of Lamiaceae and is known for its various health-promoting properties [9,10].

The enzymatic activity of a number of oxygenases, reductases and transferases gives rise to the complex and heterogenous metabolic profiles of phenylpropanoids [11,12]. Rosmarinic acid biosynthesis of RA has been elucidated, including enzymes catalyzing consecutive steps and genes encoding them in such Lamiaceae species as *Coleus blumei* Benth. [13,14,15,16], *Melissa officinalis* L. [17,18], *Agastache rugosa* (Fisch. & C.A.Mey.) Kuntze [19,20] and *Salvia miltiorrhiza* [21,22,23,24,25,26,27,28]. The full-length cDNA sequences encoding enzymes participating in RA biosynthesis from RA-synthesizing plants have been listed by Petersen [2]. Up to date, genes encoding for enzymes of the phenylpropanoid pathway from species of *Perovskia* subgenus have not been amplified and investigated.

The proposed phenolic acid biosynthetic pathway in *S. miltiorrhiza*, the closest homolog of *Perovskia* subgenus of which the phenylpropanoid biosynthesis pathway is known, is shown in Figure 1. Both the general phenylpropanoid pathway and a tyrosine-derived pathway are involved [9]. The key enzymes in the phenylpropanoid pathway include phenylalanine ammonia lyase (PAL), cinnamate 4-hydroxylase (C4H) and 4-coumarate:CoA ligase (4CL). In the tyrosine-derived pathway, tyrosine aminotransferase (TAT) and 4-hydroxyphenylpyruvate reductase (HPPR) are active.

In *S. miltiorrhiza*, a biochemical study suggested that the initial committed step towards RA involves the hydroxylation of 4-hydroxyphenyllactate (pHPL) at C-3 in the aromatic ring to generate 3,4-dihydroxyphenyllactate (DHPL) [29]. Then, rosmarinic acid synthase (RAS) couples DHPL with the 4-coumaroyl moiety to form an ester, which is hydroxylated by a cytochrome P450-dependent monooxygenase CYP98A14 to form RA. It is partially different than in other plant species, where pHPL is the direct RAS substrate, coupled with 4-coumaroyl-CoA, and RA is formed by dihydroxylation of the ester [6,9,13,18].

RA is also considered to be a precursor of biosynthesis of salvianolic acid B (SAB), which is a structural dimer of RA, but the enzyme involved in catalysis is still unknown [27]. SAB, together with the salvianolic acid E and feruloyl-3′,4′-hydroxyphenyllactic acid, are considered the end products of the phenolic acid pathway in *S. miltiorrhiza* [29]. Further important derivatives of caffeic acid found in *S. miltiorrhiza* are, among others, salvianolic acid A (SAA) and a phenolic glycoside, salviaflaside [7,10].

The subgenus *Perovskia* Kar. of the extended genus of *Salvia* comprises several perennial, shrubby, aromatic species with medicinal potential [31]. Among them, *Salvia abrotanoides* Karel. (previously known as *Perovskia abrotanoides* Kar.) and *Salvia yangii* B.T. Drew (formerly *Perovskia atriplicifolia* Benth.) are the most widespread as distributed over vast areas from Western Iran and Pakistan and as far as Tibet and Xinjiang in China [32,33,34,35].

Both *S. yangii* and *S. abrotanoides* are used as folk medicinal herbs in the areas of their natural occurrence. *S. yangii* is used as a cooling medicine in the treatment of fever, dysentery, scabies, diabetes, to heal wounds and as an antibacterial remedy [36,37,38,39,40]. In traditional Tibetan and Chinese medicine, it has been claimed that *S. yangii* is a powerful analgesic and parasiticide agent [41,42]. *S. abrotanoides*, with the common Persian name of “Brazambol,” is locally used for the treatment of typhoid, fever, headache, toothache, gonorrhoea, vomiting, cardiovascular diseases, liver fibrosis, painful urination and cough and as a sedative, analgesic and antiseptic drug [37,43,44,45,46]. *S. abrotanoides* was also proved to have cytotoxic, antiplasmodial and antiinflamatory pharmacological properties, which have been used to treat leishmaniasis in Iranian folk medicine for centuries [47,48].

Species of subgenus *Perovskia* contain a wide array of natural compounds belonging to different classes [39]. Most of the published data refer either to the composition and activities of essential oils [49,50,51,52,53] or to red-colored quinoid norabietanoids called tanshinones, which are present in the roots of species belonging to the subgenus *Perovskia* and are considered an alternative source of pharmacologically active compounds found in ‘Danshen’ (*Salviae miltiorrhizae rhizoma et radix*-a pharmacopoeial drug of China and Europe) [7,10,54,55]. Besides tanshinones, another important chemical class of a pharmacological value in the subgenus *Perovskia* is phenolic compounds. There are only several reports of phenolic compounds and their derivatives in *S. yangii* and *S. abrotanoides* [36,56,57,58,59], but none of them investigated their content in different plant organs or variations throughout the vegetation season.

Recently, a targeted comparison of the metabolic profiles of *S. yangii* and *S. abrotanoides* revealed differences between these two species, despite a close taxonomic relationship [60]. The analysis showed the presence of 14 different phenylpropanoids in leaves of both *S. yangii* and *S. abrotanoides*. Phenylpropanoid profiles in the roots were more diversified. In the roots of *S. yangii*, six phenylpropanoids were found, while the roots of *S. abrotanoides* contained four different phenylpropanoids. Rosmarinic acid, salviaflaside—a rosmarinic acid 3′ glucoside, 8-methylchromen-4-one and salvianolic acid L were detected in both species. Methoxytaxifolin, 2-*O*-*p*-coumaroyltartronic acid and melitric acid B were found only in *S. yangii* and not in *S. abrotanoides,* while a flavonoid glucoside, hesperidin, was found exclusively in the roots of *S. abrotanoides*.

Following our previous investigations, in this work, we have performed a quantitative analysis of phenylpropanoids in leaves and roots of *S. abrotanoides* and *S. yangii* during the vegetative season. By comparing phenylpropanoid profiles and seasonal changes of rosmarinic acid and its glucoside levels with transcriptional activity of genes encoding several phenylpropanoid biosynthetic pathway enzymes, we aimed at investigating mechanisms regulating rosmarinic acid biosynthesis in two *Salvia* species subg. *Perovskia*. Up to date, no studies of phenylpropanoid biosynthetic pathway at the transcriptional level has been performed in the *Perovskia* subgenus.

## 2. Materials and Methods

### 2.1. Plant Material

*Salvia abrotanoides* and *S. yangii* plants were grown at the experimental field in the Botanical Garden of Medicinal Plants (BGMP) at the Wroclaw Medical University (Poland) and as described in the previous paper [60]. In the present experiments, we used the roots (SA R and SY R) and leaves (SA L and SY L) of the wild-type clones for which the voucher specimens were deposited in the BGMP’s herbarium under the reference numbers: *S. abrotanoides*: P-122; *S. yangii*: P-123.

The harvest of leaves and roots was performed in mid-May, mid-August and mid-October 2016, which represented three growth stages during the vegetative season: the start of the season (SOS), the flowering time at the middle of the season (MOS) and the end of the season (EOS). Meteorological data for this location were monitored by the State Institute of Meteorology and Water Resources (IMGW, Wroclaw, Poland) Station in Wroclaw city (located 500 m from the BGMP). Selected climate features of the year of harvest (2016) together with the data of the precedent years 2015, 2014 and the period of 1986–2005 are shown in the Appendix A. The plant material for both molecular and phytochemical analysis was processed as described previously [60].

### 2.2. Sample Preparation for Qualitative Analysis

The solvents and other chemicals for extraction and chromatography were purchased from Merck (Darmstadt, Germany) and were of analytical or LC-MS grade.

The samples for qualitative profiling were prepared according to the procedure published previously [60].

### 2.3. Liquid Chromatography–Mass Spectrometry Analysis

Ultra-high performance liquid chromatography quadrupole time of flight mass spectrometry (UHPLC-QTOF-MS) was carried out using Thermo Ultimate 3000 RS (Thermo Fisher Scientific, Waltham, MA, USA) chromatographic system coupled to a Bruker Compact (Bruker, Billerica, MA, USA) quadrupole time of flight (QTOF) mass spectrometer. 

The detailed system settings and chromatography-mass spectrometry analysis conditions have been described previously [60].

Quantitative analysis using DAD-HPLC was performed using external standards of rosmarinic acid (CAS 537-15-5 (R)-rosmarinic acid, EDQM, Strasbourg, Europe) and salviaflaside (CAS 178895-25-5, purity > 99% by HPLC, isolated in-house from the roots of *S. yangii*, and identified using NMR spectroscopy, Appendix A).

Calibration curves were constructed from 7-point linear concentration range from 800 to 3.9 µg/mL (r^2^ ≥ 0.9998) for rosmarinic acid and 1300 to 10.1 µg/mL (r^2^ ≥ 0.9998) for salviaflaside. Each of the 7calibration levels was analyzed 3 times. Quantification was conducted based on peak areas. The concentration of compounds was expressed in milligram per gram dry weight (mg/g). Since the salvianolic acid L is not available commercially, quantification of salvianolic acid L was conducted by using a rosmarinic acid calibration curve, and the results are shown in the Appendix A. 

### 2.4. Amplification of Partial Cds of Enzymes from Phenylpropanoid Biosynthesis Pathway in S. yangii and S. abrotanoides

Primers for amplification of genes of the rosmarinic acid biosynthesis pathway (Table 1) were designed based on homologies found in genes isolated and sequenced earlier. *TAT* primers were designed using KM575934.1, DQ334606.1, KM053278.1, HQ221576.1 and MT268329.1; *HPPR* primers were designed based on: KC834747.1, EU924744.1, JX566894.1, KM591599.1, DQ266514.1 and HM587131.1; *PAL* primers were designed using: DQ408636.1, AF326116.1, KF220569.1, KJ010815.1, JQ277717.1, FN665700.1 and EF462460.1; *C4H* primers were designed with: DQ355979.1, KM434189.1, MH208308.2, MH208306.2, MH208309.2 and MH208307.2; *4CL* primers were designed using: AY237164.1, FN665699.1, MN599456.1, KJ010817.1 and KF643242.1; *RAS1* primers were designed based on: KM575933.1 and FJ906696.1; for *RAS2* primers KF220573.1 and KF220571.1 were used; *Cyp98A14* primers were designed based on HQ316179.1 and KP337738.1.

PCR reactions were performed in a T-100 thermal cycler (BIO-RAD, USA) in 20 μL volumes containing template DNA and were set up according to the Q5™ High-Fidelity DNA Polymerase manufacturer protocol (Q5™ High-Fidelity DNA Polymerase, New England BioLabs Ltd., Hitchin, UK). PCR products were visualized by gel electrophoresis in 2% agarose gel in TEA buffer with 50 bp DNA Ladder (New England Biolabs Ltd.) and stained with SimplySafe (EURx, Gdansk, Poland). Amplified DNA products were purified with the Syngen Gel/PCR Mini Kit (Syngen, Wroclaw, Poland). 

### 2.5. Sanger Sequencing and BLAST Analysis

Sanger sequencing was carried out with the BrilliantDye™ Terminator v3.1 Kit (Nimagen B.V., Nijmegen, The Netherlands). Reactions were set with a 4-fold dilution of the reaction premix and the addition of BrilliantDye^®^ Terminator 5X Sequencing Buffer (Nimagen B.V., Nijmegen, The Netherlands) according to the producer’s instructions. Sequencing products were precipitated with ethanol, dissolved in TSR (Hi-Di Formamide) (Thermo Fisher Scientific, Waltham, MA, USA) and then separated by capillary electrophoresis on the Applied Biosystems™ 310 Genetic Analyzer (Thermo Fisher Scientific, Waltham, MA, USA). Capillary electrophoresis was performed in a 50 cm long capillary filled with POP-7 Polymer (Thermo Fisher Scientific, Waltham, MA, USA). The separation time was 1 h 40 min, and the run voltage was 6 kV with 60 sec 2 kV sample injection. Two reads were collected for each sample. Sixteen sequences were generated for molecular analysis, eight cds for *S. abrotanoides* (MN248746.1, MN248742.1, MN248744.1, MN248740.1, MN248738.1, MW854244.1, MW854240.1 and MW854242.1) and eight for *S. yangii* (MN248747.1, MN248743.1, MN248745.1, MN248741.1, MN248739.1, MW854245.1, MW854241.1 and MW854243.1).

The sequence quality was checked with the Sanger Quality Check App (Thermo Fisher Scientific, Waltham, MA, USA). Forward and reverse sequencing reads for each cds were assembled into a contig using the BioEdit Sequence Alignment Editor [61]. The identity of all obtained sequences was confirmed through similarity to published sequences using the BLAST algorithm.

### 2.6. RNA Extraction and cDNA Synthesis

Total RNA from powdered leave material was extracted with the Plant/Fungi Total RNA Purification Kit (Norgen Biotek Corp., Thorold, Canada) and treated with the Turbo DNA-*free*^™^ (Thermo Fisher Scientific, Waltham, MA, USA). Prior to and after the DNase treatment, the concentration and quality of total RNA were assessed by a NanoDrop^™^ 2000c spectrophotometer (Thermo Fisher Scientific, Waltham, MA, USA), while the integrity was checked by the 2100 Bioanalyzer system (Agilent Technologies, Santa Clara, CA, USA).

Single-strand cDNA was synthesized from DNase-treated RNA using SuperScript^®^ IV (Thermo Fisher Scientific, Waltham, MA, USA) reverse transcriptase with oligo(dT)_18_ primer (Meridian Bioscience, Cincinnati, OH, USA). The assessment of cDNA quantity and quality was performed both spectrophotometrically by the NanoDrop ^™^ 2000c and in real-time PCR.

### 2.7. Expression Analysis of the Phenylpropanoid Pathway Genes

The gene-specific primers listed in Table 1 were used for the analysis of gene expression based on reverse-transcribed cDNA products as templates. The real-time PCR primers for *PAL*, *C4H*, *4CL*, *TAT*, *HPPR*, *RAS1*, *RAS2* and *Cyp98A14* in *S. yangii* and *S. abrotanoides* were designed with Primer3Plus (http://www.bioinformatics.nl/primer3plus, accessed on 31 July 2019 and 24 April 2021) using the following sequences of: *TAT* (accession numbers: MN248746.1 and MN248747.1), *HPPR*, (accession numbers: MN248742.1 and MN248743.1); *PAL* (accession numbers: MN248744.1 and MN248745.1); *C4H* (accession numbers: MN248740.1 and MN248741.1), *4CL* (accession numbers: MN248738.1 and MN248739.1), *RAS1* (accession numbers: MW854244.1 and MW854245.1), *RAS2* (accession numbers: MW854240.1 and MW854241.1) and *Cyp98A14* (accession numbers: MW854242.1 and MW854243.1) [62]. Each real-time PCR reaction was performed in triplicates. The instrumental settings and quantitative Real-time PCR conditions were used as previously published [20]. The relative quantification using *actin* (*ACT*) genes (accession numbers MW240684 and MW240685) as a reference was used for the determination of the transcript levels. Reaction efficiency for each pair of primers was estimated experimentally (by calibration curve) and used for calculations using the Pfaffl mathematical model [63]. All steps starting from experiment design through RNA isolation to data analysis were conducted according to the MIQE guidelines provided for quantitative real-time PCR analysis [64]. Normalized expression (NE) data are presented in Appendix A. 

### 2.8. Statistical Data Evaluation

A one-way ANOVA with post hoc Tukey’s multiple comparison tests (GraphPad Prism, San Diego, CA, USA) was used to determine the statistical significance of differences between phytochemical profiles and, in parallel, between means of expression data sets. Differences were regarded as significant at *p* < 0.05. (Appendix A). 

Pearson coefficient analysis (*p* < 0.05) was performed using Statistica 13.3PL (StatSoft, Krakow, Poland). Species-adjusted partial correlation analyses were performed to determine the association between metabolite levels with gene expression levels in leaves and roots of *S. yangii* and *S. abrotanoides* at different time intervals.

## 3. Results

### 3.1. Analysis of Phenolic Compounds

Out of 14 different phenylpropanoids detected earlier in the leaves of both *S. yangii* and *S. abrotanoides* [60], rosmarinic acid was estimated quantitatively. In the roots of *S. yangii*, six different phenylpropanoids were found by UHPLC-QTOF-MS analysis, while the roots of *S. abrotanoides* contained four different phenylpropanoids [60]. Out of these, two phenolic compounds were estimated quantitatively, namely: rosmarinic acid and salviaflaside (a rosmarinic acid 3′-glucoside), whereas salvianolic acid L (a tetrameric molecule in which the condensation of caffeic and dihydroxyphenyllactic acid side chains forms a naphthalene motif) was estimated based on rosmarinic acid standard (Figure 2).

The number of phenolic compounds differed between species and between organs (Table 2). When analyzed in the course of the vegetative season, a significant difference between the level of RA in leaves of *S. abrotanoides* (35.17 mg/g d.w.) and *S. yangii* (88.64 mg/g d.w.) was found at the beginning of the season (SOS). During flowering (MOS), the difference was less pronounced (24.66 mg/g d.w. in *S. abrotanoides* and 39.31 mg/g d.w. in *S. yangii*), and at the end of the season (EOS), the level of RA in leaves became nearly equal yielding 58 mg/g d.w. and 53.84 mg/g d.w. in *S. abrotanoides* and *S. yangii*, respectively.

Quantitative analysis showed that, in roots, the profile of phenolic compounds was clearly dominated by rosmarinic acid. Unlike in the leaves, in the roots, the amount of RA was significantly higher in *S. abrotanoides* than in *S. yangii* throughout the vegetative season (Table 2). The highest level of RA was noted in the roots of *S. abrotanoides* (65.56 mg/g d.w.) at the beginning of the season (SOS), while the lowest RA amount was found in the roots of *S. yangii* (19.85 mg/g d.w.) at the end of the season (EOS). Additionally, the roots of *S. abrotanoides* showed a considerable amount of salviaflaside, which accumulated at the end of vegetation season (EOS), reaching 4.70 mg/g d.w. in *S. abrotanoides* and 3.08 mg/g d.w. in *S. yangii*. Salvianolic acid L turned out to be the least abundant of all phenolic compounds found in roots, reaching 0.99 mg/g d.w. rosmarinic acid equivalents in flowering *S. yangii* (MOS), while in *S. abrotanoides,* it varied from 0.5 mg/g d.w. rosmarinic acid equivalents (SOS) to an undetectable level in EOS (Appendix A).

When comparing between organs, significantly higher amounts of RA were found in roots than in leaves of *S. abrotanoides* at the beginning and in the middle of the season (SOS, MOS) (Table 2). In *S. yangii*, RA accumulated in leaves, where there was a significantly higher amount, compared to roots, was found at the beginning of the season (SOS), and this tendency was kept throughout the whole season. 

### 3.2. BLAST Analysis Results of Genes Encoding Phenylpropanoid Biosynthetic Enzymes from S. abrotanoides and S. yangii

The identity of all obtained sequences was confirmed through similarity to published sequences using the BLAST algorithm. DNA sequences obtained for *TAT*, *PAL*, *C4H*, *4CL*, *RAS1*, *RAS2* and *Cyp98A14* revealed the highest homology with corresponding sequences from *Salvia miltiorrhiza*, while *HPPR* sequences shared the highest sequence similarity with the corresponding sequence from *Salvia officinalis*. (Table 3). Apart from that, the BLAST analysis showed very high mutual sequence similarity of gene sequences obtained from *S. abrotanoides* and *S. yangii*, which, however, was not included in Table 3 for better clarity.

### 3.3. Expression Levels of Genes Involved in Biosynthesis of Phenylpropanoids

Quantitative real-time PCR analysis was used to investigate the expression pattern of *PAL*, *C4H*, *4CL*, *TAT*, *HPPR*, *RAS1*, *RAS2* and *Cyp98A14* in leaves and roots of *S. abrotanoides* and *S. yangii*. Gene transcript levels in each sample were normalized to the respective transcript level of *ACT,* which generated a normalized expression value (NE) (Figure 3). This analysis showed that, in leaves, the expression of the majority of genes encoding general phenylpropanoid pathway enzymes, a tyrosine-derived pathway, *RAS1* and *Cyp98A14* were significantly higher at the beginning of the season (SOS) than in two others analyzed time points. Indeed, the highest transcript levels have been detected for *C4H* (12.37), *TAT* (8.6) and *PAL* (8.5) in leaves of *S. abrotanoides* at the beginning of the season. In roots, transcript levels of all genes were the lowest at the end of the analyzed period (EOS).

When comparing transcript levels between species, all genes exhibited lower expression in *S. yangii* than in *S. abrotanoides* (Figure 3), with a majority of these differences being statistically significant at *p* < 0.05. This phenomenon was particularly noticeable at the beginning of the season (SOS) in leaves and roots of *S. yangii* for *PAL*, *C4H*, *TAT* and *RAS1*, for *HPPR* in leaves and for *4CL* in roots. In the middle of the season (MOS), significantly lower transcripts levels in *S. yangii* were noted for *RAS2* in leaves and in most other analyzed genes in roots. At the end of the season, no significant differences in transcript levels of analyzed genes, except for *RAS1*, were observed. 

The comparison between roots and leaves of the same species revealed significantly lower transcript levels of *HPPR*, *RAS1*, *RAS2* in roots than in leaves in two or three analyzed time points (Figure 3). *PAL*, *C4H* and *Cyp98A14* exhibited lower expression in roots than in leaves at the beginning of the season (SOS). In *S. abrotanoides*, the transcript levels of *PAL*, *C4H*, *4CL* and *TAT* were significantly higher in roots (dark grey bars in Figure 3) than in leaves (light grey bars) in the middle of the season (MOS).

Two gene isoforms of RAS, which has been amplified in *S. abrotanoides* and *S. yangii*, *RAS1* and *RAS2*, displayed different tissue specificity and expression pattern (Figure 3). In leaves and roots of both species, the *RAS1* gene presented a significantly higher expression level than *RAS2*, which suggests that RAS1 is the main isoform active in both species. RAS2 turned out to be the leaf-specific isoform as it was expressed in leaves and barely in roots (at the limit of detection). Similar to most of the other genes from the phenylpropanoid biosynthesis pathway, *RAS1* showed the highest expression at the beginning of the analyzed vegetation period (SOS), which significantly decreased in time in leaves of both species. The decreasing tendency of *RAS1* transcript level was also visible throughout the analyzed period in roots; however, the differences in its abundance were not significant. 

### 3.4. Partial Correlation Analysis between Metabolite Levels and Biosynthetic Pathway Gene Expression

Species-adjusted partial correlation analyses were performed to determine the association between metabolite levels with gene expression levels in leaves and roots of *S. yangii* and *S. abrotanoides* at different time intervals (Table 4). A significant positive partial correlation was obtained between rosmarinic acid content in roots and expression levels of all analyzed genes from the phenylpropanoid biosynthesis pathway, except for the *RAS2*. A significant negative partial correlation of salviaflaside content was noted with the expression of *C4H*, *4CL*, *TAT*, *HPPR* and *RAS1* in roots. 

## 4. Discussion

This work continues our previous investigations in which we have provided a comprehensive analysis of the phytochemical profile of non-volatile compounds, performed separately for roots and leaves of field-cultivated *S. yangii* and *S. abrotanoides* [60]. Targeted comparison of metabolic profiles of *S. yangii* and *S. abrotanoides* revealed important differences in the profile of non-volatile specialized metabolites, including phenolic acids. In this work, we have performed quantitative analysis of phenylpropanoids in leaves and roots of *S. abrotanoides* and *S. yangii* in the course of the vegetative season. Out of 14 (in leaves) and 6 or 4 (in roots) phenolic compounds detected earlier, we were able to quantify rosmarinic acid in leaves and roots as well as salviaflaside found in roots of both analyzed species. Although the phenylpropanoid profile appeared to be clearly dominated by RA in both *S. abrotanoides* and *S. yangii*, the number of phenolic compounds differed between species and between organs throughout the analyzed vegetative period (Table 3).

Phenylpropanoid profile of species in subg. *Perovskia* turned out to be less complex than in other species within the Lamiaceae family. The profile is clearly dominated by RA. Unlike in other Lamiaceae species such as *S. milthiorrhiza* [65,66] or *Agastache rugosa* [19,20], no compounds from the flavonoid class were detected. 

Rosmarinic acid is a well-known, common phytochemical constituent of Boraginaceae and Lamiaceae, in which it is regarded as a typical product of the Nepetoideae subfamily [9]. However, as RA has also been described from other plant families such as ferns of the family Blechnaceae, lower plants such as the hornworts and in monocotyledonous plants such as the sea grass family Zosteraceae, it cannot be used as a chemotaxonomical marker to differentiate among families [6,9]. 

Numerous studies performed to date have shown various levels of RA across different plants from the Lamiaceae family [5,67,68]. Seasonal variations in RA levels have been followed in *Rosmarinus officinalis* (now *S. rosmarinus*) [69,70,71], *Thymus longicaulis* [72] and *Perilla frutescens* [73]. The profile of RA accumulation differed between plants cultivated in glasshouses and collected in the wild and most probably depended on the bioclimatic area, latitude and plant’s growth phase. In the glasshouse-cultivated rosemary plants, the RA content was found to be constant over a period of one year under southern UK conditions [70], while glasshouse-cultivated plants in Spain showed the highest amounts of RA in February [69]. Although several reports of phenolic compounds and their derivatives in *S. yangii* and *S. abrotanoides* have been released to date [36,56,57,58,59], none of them investigated their content in different plant organs or variations throughout the vegetation season.

In this work, we have performed qualitative observations of phenylpropanoid levels in separate roots and leaves of *S. yangii* and *S. abrotanoides* cultivated in Europe (Wroclaw, Poland), a low altitude location with moderate cool climate with relatively high humidity and precipitation due to the Atlantic climate influences. However, a reliable comparison was possible due to ensuring the same cultivation conditions such as latitude, weather conditions, soil etc., the same sampling method (immediate freezing in liquid nitrogen), the same extraction procedure (MeOH: H_2_O; 8:2 *v/v*) and method for chemical analysis (UHPLC-QTOF-MS) as well as the same time of harvest. As we investigated field-grown perennial plants, the harvest of leaves was possible during a vegetative season. The harvest of leaves and roots was performed at three growth stages during the vegetative season: at the start of the season (SOS), during flowering at the middle of the season (MOS), and at the end of the season (EOS).

To enable quantitative analysis of transcript levels of genes encoding various phenylpropanoid biosynthetic pathway enzymes, we amplified and sequenced 16 partial cds of genes encoding for *PAL*, *C4H*, *4CL*, *TAT*, *HPPR*, *RAS1*, *RAS2* and *Cyp98A14*; 8 sequences were amplified from *S. abrotanoides* and another 8 from *S. yangii* (Table 3). To our knowledge, this is the first report of sequencing genes encoding for enzymes of the phenylpropanoid pathway from species of *Perovskia* subgenus. This enabled further quantitative real-time PCR analysis to investigate the expression pattern of *PAL*, *C4H*, *4CL*, *TAT*, *HPPR*, *RAS1*, *RAS2* and *Cyp98A14* in leaves and roots of *S. abrotanoides* and *S. yangii* (Figure 3). This analysis revealed that the expression of the majority of studied genes was high at the beginning of the season and was decreasing over the analyzed period. This was true for all genes (except for *RAS2*) in leaves, while in roots, relatively high levels of transcripts were found at the beginning and in the middle of the season. When comparing transcript levels between species, many genes exhibited significantly lower expression in *S. yangii* than in *S. abrotanoides*. Comparison between roots and leaves of the same species revealed significantly lower transcript levels of the majority of genes in roots than in leaves, in at least two analyzed timepoints. Gene expression analysis of the phenolic acids biosynthesis-related genes in *S. miltiorrhiza* has been reported previously [26,30], no such studies were carried out for species of *Perovskia* subgenus to date.

Simultaneous analysis of both phenylpropanoid profiles and transcript levels of genes from respective biosynthetic pathways performed over the course of the vegetative season has served as a model to observe possible relationships between transcript and metabolite accumulation patterns. By employing species-adjusted partial correlation analyses, we were able to determine the association between metabolite levels with gene expression levels in leaves and roots of *S. yangii* and *S. abrotanoides* at different time intervals (Table 4). Interestingly, in roots, a significant positive partial correlation was obtained between rosmarinic acid content and expression levels of all analyzed genes from the phenylpropanoid biosynthesis pathway, except for the *RAS2* (which is barely expressed in roots). This suggests that genes encoding for *PAL*, *C4H*, *4CL*, *TAT*, *HPPR*, *Cyp98A14* and, among two *RAS* isoforms, the *RAS1* isoform, which was amplified in this work, are probably the main isoforms directly involved in the biosynthesis of RA in roots of *S. yangii* and *S. abrotanoides*. The positive correlation between RA and respective gene expression levels suggests, in field-grown *S. yangii* and *S. abrotanoides,* the biosynthesis of RA in roots might be regulated at the transcriptional level by a positive feedback mechanism, in which high gene transcript levels translate into high RA content. A highly significant correlation was obtained between RA levels and transcript levels of *RAS1* and *Cyp98A14*—genes encoding two enzymes acting downstream the condensation reaction and leading directly to the synthesis of RA. This suggests that RAS1 and Cyp98A14 might be key and rate-limiting enzymes in the RA biosynthesis pathway in roots.

No correlation between rosmarinic acid content and gene expression levels was, however, detected in leaves. A possible explanation may be based on the observation of *RAS* gene expression in leaves. Both isoforms of *RAS* amplified in *S. yangii* and *S. abrotanoides* turned out to be active in leaves (Figure 3), which probably means that they both contribute to the final level of RA in this organ. As RA accumulation in leaves presumably results from the net activity of *RAS1* and *RAS2* (whether coordinated or not), the direct correlation between RA content with neither of the *RAS* isoforms is visible. At present, nothing is known about the enzymatic properties of these isoforms, such as specificity, efficiency and regulation of activity by physical conditions, all of which can add to the sum of the final product in the plant material. However, the possibility of long-distance transport of either precursors or the final products cannot be excluded. Very little is known about long-distance (or inter-organ) transport of plant phenolics, with only scarce evidence for such moves of flavonoids and salicylic acid obtained from model species [74,75,76]. However, none of these hypothetical mechanisms were possible to be verified using the present methodology. 

A different rate of accumulation and/or degradation of certain metabolites (such as RA) could also lead to variations in their endpoint levels, independently from key biosynthetic gene activities. Future studies are highly needed to elucidate the metabolic fluxes that influence the statically observed phytochemical profiles.

Is it also possible that *S. yangii* and *S. abrotanoides* possess other isoforms of enzymes from the general phenylpropanoid pathway, the tyrosine-derived pathway and *Cyp98A14*; we were unable to confirm the presence through the degenerated-primers amplification approach. In fact, it has been demonstrated that enzyme-encoding genes in the phenylpropanoid pathway are present as small gene families in the model plant *Arabidopsis thaliana* [77,78,79,80]. The *PAL* gene family comprises 1–5 genes in other species, such as *Populus tremuloides* [81], *Populus trichocarpa* [82], *Coffea canephora* [83], and *Nicotiana tabacum* [84]. Hence, the differences in the phenylpropanoid pathway biochemical output may appear as a result of sequence and copy number variation. The extra copies or gene family members may undergo neofunctionalization, differential expression or assembly of the proteins in metabolon-type complexes, leading to the formation of divergent metabolic profiles. For example, the *Populus tremuloides* PAL isoforms participate either in condensed tannin (*PtPAL1)* or lignin (*PtPAL2*) biosynthesis [81]. Elucidating the taxon-specific characteristics of phenolic acid metabolism requires extensive analysis of all duplicated gene copies involved.

In *S. miltiorrhiza*, gene families (of a various number of genes, depending on a study) encoding enzymes involved in phenolic acids biosynthesis were found through several RNA-sequencing approaches [28,85,86]. A genome-wide study using the new generation sequencing technology would certainly help revealing more whole-gene sequences for respective gene families in species of *Perovskia* subgenus, as it was completed for *S. miltiorrhiza*. 

Due to the degradation of genetic resources and variable content of phenolic acids in traditionally used *S. miltiorrhiza*, limited phenolic acid production cannot meet the increasing market demand [7]. *S. yangii* and *S. abrotanoides* could serve as an alternative source for pharmaceutically important phenolic acids; however, harvesting medicinally important plants from the wild makes them critically endangered and affects the environmental biodiversity. Thus, it is extremely important to use modern biotechnology methods and precision agriculture to increase the yield of phenolic acids in plants. Apart from various biological methods, such as culturing hairy roots, callus, suspension cells, and using elicitors treatment, transcriptional regulation and genetic engineering could be employed for increasing the production of phenolic acids in these plants. 

Therefore, understanding the factors and mechanisms regulating the accumulation of secondary metabolites is essential for future manipulation of their qualitative and quantitative production in plants. Many previous studies documented the influence of various factors on phytochemical profiles in medicinal and aromatic plants. The environmental factors change along the season, for example, causing periodical water deficits in summer that may lead to metabolic adjustments via associated gene expression modulation [87,88,89,90].

In this work, we have identified 16 gene candidates with properties correlating to phenolic acid biosynthesis. These were *PAL*, *C4H*, *4CL*, *TAT*, *HPPR*, *RAS1*, *RAS2* and *Cyp98A14*, each in both *S. yangii* and *S. abrotanoides*. By comparing phenylpropanoid profiles with respective gene transcript levels, we have observed the positive feedback mechanism regulating the biosynthesis of RA in roots of *S. yangii* and *S. abrotanoides*, acting at the transcriptional level. Additionally, our study offered elucidation of rate-limiting steps regulating phenylpropanoid biosynthesis on a transcription level in these plants. By performing partial correlation analysis, we have identified *RAS1* and *Cyp98A14* as putative functional candidates for further characterization via reverse genetic approaches to practically manipulate rosmarinic acid production. In future, this will facilitate the gene-based metabolic enhancement of phenolic compounds production in these important and prospective medicinal herbs.

## 5. Conclusions

The comparative analysis of the phenolic profile and key phenylpropanoid biosynthesis genes transcriptional activity in two closely related medicinal species of *Perovskia* during the vegetative season, eight genes correlating to the main compound, rosmarinic acid, were pointed out (*PAL*, *C4H*, *4CL*, *TAT*, *HPPR*, *RAS1*, *RAS2* and *Cyp98A14*, each in both *S. yangii* and *S. abrotanoides*). Furthermore, we identified *RAS1* and *Cyp98A14* as potential targets for functional manipulation of caffeic acid oligomers biosynthesis.

The high correlation in the roots and lack of correlation in the leaves in both species suggests that the biosynthesis of RA in roots is regulated at the transcriptional level by a positive feedback mechanism, in which high gene transcript levels translate into high RA content. Conversely, unknown mechanisms are supposed to exist in the leaves that might include a complex orchestrating of RAS1 and RAS2 isoforms activities and/or transcription-independent mechanisms such as long-distance transport or varied metabolic fluxes in the form of downstream biosynthetic steps or degradation of rosmarinic acid.

The observed seasonal differences are also essential for the rational use of these underutilized plants as sources of bioactive natural products.

## Figures and Tables

**Figure 1 biology-10-00458-f001:**
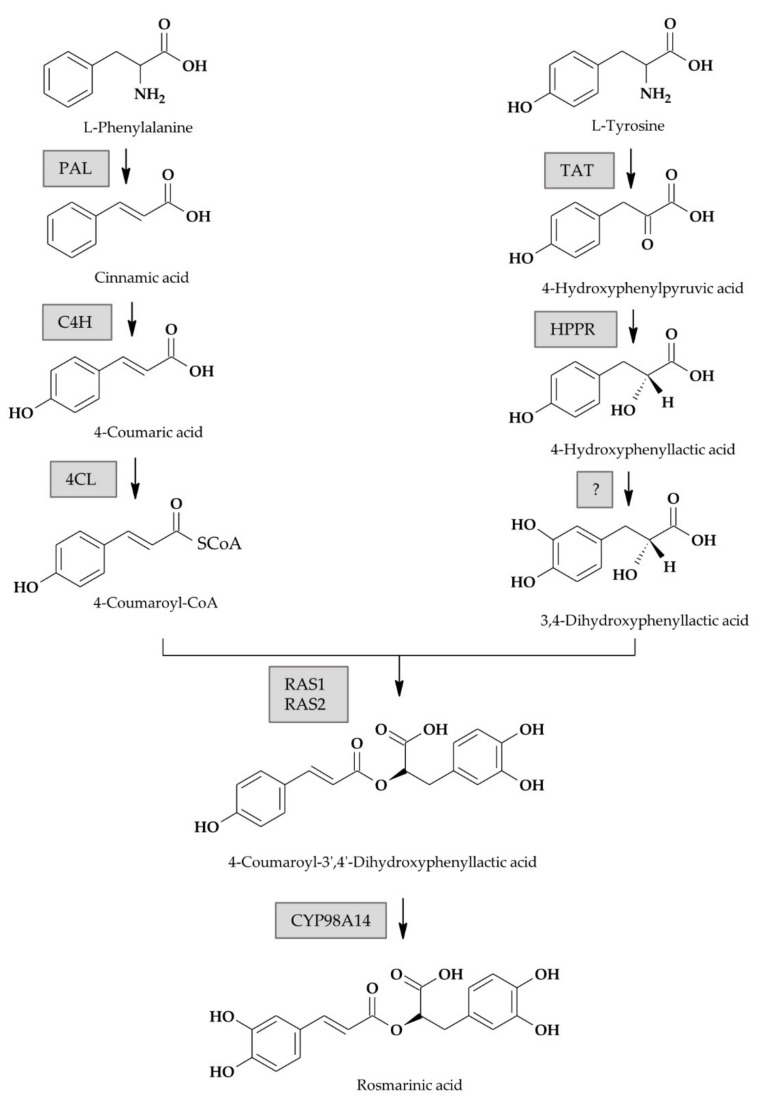
Putative biosynthetic pathway of rosmarinic acid according to previous studies on *S. miltiorrhiza*, a close relative of *S. yangii* and *S. abrotanoides* [7,26,27,29,30] Framed are genes investigated in this study. (*PAL*, phenylalanine ammonia lyase; *C4H*, cinnamate 4-hydroxylase; *4CL*, 4-coumaryl-CoA ligase; *TAT*, tyrosine aminotransferase; *HPPR*, hydroxyphenylpyruvate reductase; *RAS1* and *RAS2*, two isoforms of rosmarinic acid synthase; Cyp98A14, cytochrome P450-dependent monooxygenase).

**Figure 2 biology-10-00458-f002:**
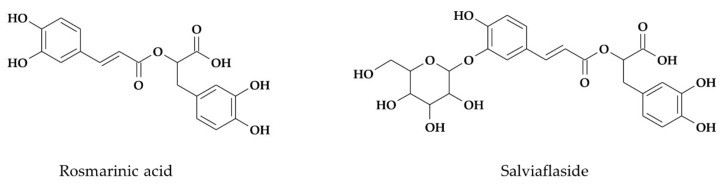
Chemical structures of phenolic compounds determined quantitatively in *S. yangii* and *S. abrotanoides*.

**Figure 3 biology-10-00458-f003:**
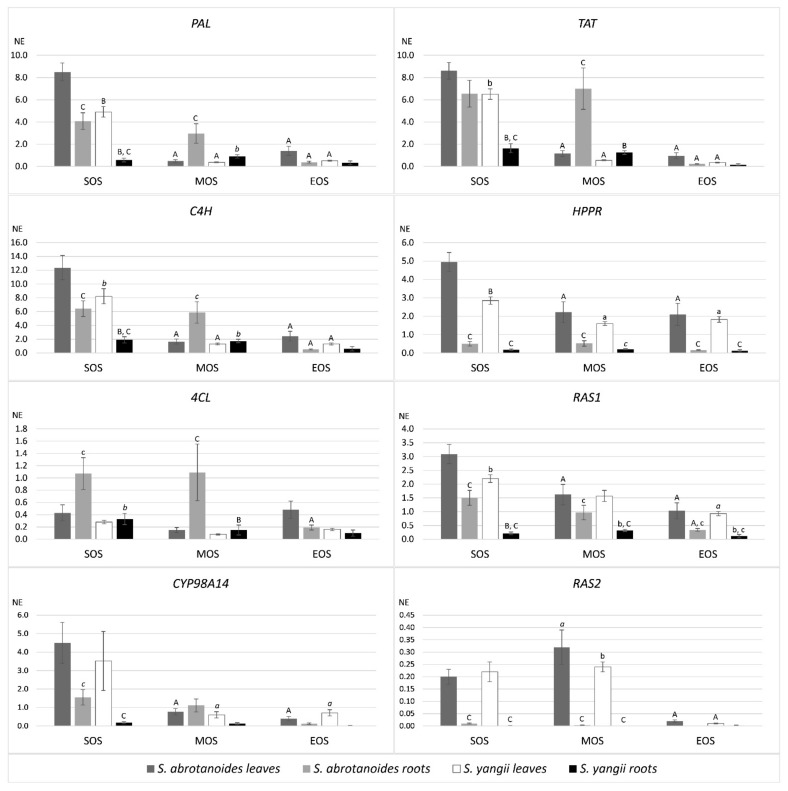
Expression levels of rosmarinic acid biosynthesis pathway genes in leaves and roots of *S. abrotanoides* and *S. yangii* in three growth stages during the vegetative season (SOS—the start of the season, MOS—the middle of the season and EOS—the end of the season.). The bars represent transcript levels normalized to *ACT*. NE, normalized expression. The letters above the error bars refer to statistically significant differences as follows: a—significant differences in comparison to SOS, b—significant differences in comparison to *S. abrotanoides*, c—significant differences in comparison to leaves. The level of difference significance is indicated by the used fonts: *p* < 0.05 lower case font, *p* < 0.001 italic font, *p* < 0.0001 upper case font.

**Table 1 biology-10-00458-t001:** Primers designed and used in this study.

Primer Name	Primer Sequence (5’ to 3’)
use: partial sequence	
TAT_F	CBATTAAGGGGATTTTGGGGYT
TAT_R	GCRAKYCCWGGAAGAATKAT
HPPR_F	BGTTYTSATGATGTGCCCBA
HPPR_R	GGCTTGCYGGMRAAGTGWGC
PALp1_F	YGAYTGGGTYATGGASAGCA
PALp1_R	CCRATRGGRGTDCCYTGGAA
PALp2_F	CYAAGATGATCGAGMGRGAG
PALp2_R	AAMACCYTVTCRCACTCYTC
C4H_F	ATGGRGAGAGRAGCMGATTG
C4H_R	AATGTGSAGRCTGAACTG
C4H_R	CAAAATGTGSAGRCTGAACTG
4CL_F	SGTGATGCTSACGCACAA
4CL_F	ACACGTACGAGGAGGTSGAG
4CL_R	GGCATCKATGAAYCCRATGT
RAS1_F	TCAAAGCCAAGTGCAAAGCC
RAS1_R	CTTCTCAAAGCGCTCCATGTG
RAS2_F	AATCCGGCTACACCACGTAC
RAS2_R	TGAAAGGCCTCCATATGCGG
Cyp98A14_F	ACCCTCAAGGACGAGTACGA
Cyp98A14_R	CATCGACGTGGTGAGGTCAA
use: real-time PCR	
ACT_RTF	ACCTCAAAATAGCATGGGGAAGT
ACT_RTR	GGCCGTTCTCTCACTTTATGCTA
TAT_RTF	ACGGACTTTGTGCCTCATTC
TAT_RTR	CATCGGCAATCACCACTATG
HPPR_RTF	GAGAGGGGCCTGGAAATTAG
HPPR_RTR	GTGGTAACTGATGGGGCAAT
PAL_RTF	ACATCCTGGCCGTCCTATC
PAL_RTR	GTCCTGCTCGTGCAGCTT
4CL_RTF	GCGATCTTGATCATGCAGAA
4CL_RTR	AAGGTCATATTTGCCCACCA
C4H_RTF	AGGAAAGGAGGTTGCAGCTT
C4H_RTR	CCCCACTCAATCGACCATAG
RAS1_RTF	TCGATTTCTTGGAGCTGCAG
RAS1_RTR	GCACCCAACTAATCACCCAAAG
RAS2_RTF	CGTGAGGTGCCCTAATTTTGG
RAS2_RTR	TTTCCGCATCAACGAAGAGC
Cyp98A14_RTF	AAACCTTCCCTACCTGCAGTG
Cyp98A14_RTR	AGCTTGACATTGGTGTTGGC

**Table 2 biology-10-00458-t002:** The content of rosmarinic acid and salviaflaside [mg/g d.w. ± SD, *n* = 3] in leaves and roots of *S. abrotanoides* and *S. yangii*, at the start of the vegetation season (SOS), middle of the season (MOS) and end of the season (EOS). Statistical significance of differences in chemical parameters between samples was evaluated with a one-way ANOVA with post hoc Tukey’s multiple comparison tests.

	*Salvia abrotanoides*			*Salvia yangii*		
Compound	SOS	MOS	EOS	SOS	MOS	EOS
	leaves					
Rosmarinic acid	35.17 ± 2.81	24.66 ± 4.16	58.00 ± 3.96 ^a^	88.64 ± 10.55 ^B^	39.31 ± 5.26 ^A^	53.84 ± 9.97 ^A^
	roots					
Rosmarinic acid	65.56 ± 5.71 ^C^	60.18 ± 8.23 ^C^	44.10 ± 0.33 ^a^	23.65 ± 2.09 ^B,C^	26.70 ± 3.76 ^B^	19.85 ± 0.87 ^b,c^
Salviaflaside	1.63 ± 0.25	0.57 ± 0.09 ^a^	4.70 ± 0.33 ^A^	0.70 ± 0.09 ^b^	1.92 ± 0.33 *^a^*^,*b*^	3.08 ± 0.11 ^A,B^

a—significant differences in comparison to SOS, b—significant differences in comparison to *S. abrotanoides*, c—significant differences in comparison to leaves, *p* < 0.05 lower case font, *p* < 0.001 italic font, *p* < 0.0001 upper case font, SOS—the start of the season, MOS—the middle of the season, EOS—the end of the season.

**Table 3 biology-10-00458-t003:** Results of the BLAST analysis.

Accession Number (Query)	Gene (Query)	Species (Query)	BLAST Result [Species; Gene]	Accession Number (Result)	Identity [%]
MN248746.1	*TAT*	*Salvia abrotanoides*	*Salvia miltiorrhiza; TAT*	KM575934.1	95.37
MN248747.1	*TAT*	*Salvia yangii*	*Salvia miltiorrhiza; TAT*	KM575934.1	95.18
MN248742.1	*HPPR*	*Salvia abrotanoides*	*Salvia officinalis; HPPR*	EU924744.1	94.37
MN248743.1	*HPPR*	*Salvia yangii*	*Salvia officinalis; HPPR*	EU924744.1	94.77
MN248744.1	*PAL*	*Salvia abrotanoides*	*Salvia miltiorrhiza; PAL*	DQ408636.1	91.28
MN248745.1	*PAL*	*Salvia yangii*	*Salvia miltiorrhiza; PAL*	DQ408636.1	91.22
MN248740.1	*C4H*	*Salvia abrotanoides*	*Salvia miltiorrhiza; C4H*	DQ355979.1	92.82
MN248741.1	*C4H*	*Salvia yangii*	*Salvia miltiorrhiza; C4H*	DQ355979.1	92.61
MN248738.1	*4CL*	*Salvia abrotanoides*	*Salvia miltiorrhiza; 4CL*	AY237164.1	91.88
MN248739.1	*4CL*	*Salvia yangii*	*Salvia miltiorrhiza; 4CL*	AY237164.1	92.71
MW854244.1	*RAS1*	*Salvia abrotanoides*	*Salvia miltiorrhiza; RAS*	FJ906696.1	90.89
MW854245.1	*RAS1*	*Salvia yangii*	*Salvia miltiorrhiza; RAS*	FJ906696.1	90.51
MW854240.1	*RAS2*	*Salvia abrotanoides*	*Salvia miltiorrhiza; RAS3*	KF220571.1	85.18
MW854241.1	*RAS2*	*Salvia yangii*	*Salvia miltiorrhiza; RAS3/RAS4*	KF220571.1/ KF220572.1	84.76/84.72
MW854242.1	*Cyp98A14*	*Salvia abrotanoides*	*Salvia miltiorrhiza; Cyp98A78*	KP337738.1	93.73
MW854243.1	*Cyp98A14*	*Salvia yangii*	*Salvia miltiorrhiza; Cyp98A78*	KP337738.1	93.55

**Table 4 biology-10-00458-t004:** Pearson’s correlation coefficient (r) analysis comparing the species-adjusted partial correlation between expression of rosmarinic acid biosynthesis pathway genes and rosmarinic acid and salviaflaside L content. * indicates a significant result at *p* < 0.05 and ** indicates a significant result at *p* < 0.01.

RA Biosynthesis Pathway Genes	Pearson Coefficient (r)		
	leaves	roots	
	Rosmarinic acid	Rosmarinic acid	Salviaflaside
*PAL*	0.35	0.99 **	−0.77
*C4H*	0.38	0.97 **	−0.90 *
*4CL*	0.81	0.92 *	−0.93 *
*TAT*	0.39	0.94 *	−0.93 *
*HPPR*	0.19	0.95 *	−0.90 *
*RAS1*	0.07	0.97 **	−0.68
*RAS2*	−0.34	0.75	−0.38
*Cyp98A14*	0.34	0.97 **	−0.79

## Data Availability

DNA sequence data generated in this study has been deposited in GeneBank at accession numbers: MN248746.1, MN248742.1, MN248744.1, MN248740.1, MN248738.1, MW854244.1, MW854240.1, MW854242.1, MN248747.1, MN248743.1, MN248745.1, MN248741.1, MN248739.1, MW854245.1, MW854241.1 and MW854243.1. The UHPLC-QTOF-MS raw data presented in this study are available on request from the corresponding author. The data are not publicly available due to the institutional policy of the authors’ employer.

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
