# Peer review of "Seasonal Variations of Rosmarinic Acid and Its Glucoside and Expression of Genes Related to Their Biosynthesis in Two Medicinal and Aromatic Species of Salvia subg. Perovskia"

_biology, 2021, doi:10.3390/biology10060458_

Round 1

Reviewer 1 Report

  1. The presentation for significant differences is not satisfying and difficult to be understood. It must be changed to the normal way that using different letters to identify there is a significant difference between means in each treatment.
  2. line 38: identified following gene candidates - identified the following gene candidates
  3. line 43: In future - In the future
  4. line 65: results from activity of - result from the activity of
  5. line 123: Beside - Besides
  6. line 348, line 388: the end of season - the end of the season
  7. line 380: all genes was - all genes were
  8. line 392: timepoints - time points
  9. line 574: of phenolic profile - of the phenolic profile
  10. line 589: for rational use - for the rational use

Author Response

The letter of response is attached as a PDF file

Reviewer 2 Report

It was a pleasure to read the manuscript of Stafiniak and colleagues. the authors propose a study that correlates the metabolomic analysis with the expression of the genes of two species belonging to the genus Salvia.

The manuscript is clear, however the results should be analyzed with an Anova-like approach. In fact, I don't think the student's t-test can be considered valid for their analyzes.
Furthermore, I suggest that the authors use the scientific name of the plant species used when they are first mentioned.

Finally, I suggest this interesting manuscript to be integrated into the bibliography

Mahdavi A, Moradi P, Mastinu A. Variation in Terpene Profiles of Thymus vulgaris in Water Deficit Stress Response. Molecules. 2020 

Author Response

(The authors gave the same response as above.)

Reviewer 3 Report

In this manuscript, the authors have tried to elucidate the correlation between the amount of three phenolic acids with transcriptional expression level of 8 proposed genes in two close Salvia species. But  I have some concerns about their experiment and conclusion.

1) In Result 3.1 Table 3, three compounds were quantified but only Rosmarinic acid had standard compound for identification and quantitation. As the other two analytes are also commercially available,  the authors should also include the other two standards for calibration curve. otherwise, the results related to the other two compounds should be removed out.

2) As the authors have done chemical profiling of all the samples from leave and root at three different stages, have you putatively identified some other important phenolic compounds?  How about their abundances? Why do you just focus on the three mentioned in the manuscript but the title is " seasonal variations of phenolic acids"? If you just focused on three (one) compounds, please precisely revise the title.

3) The authors proposed the synthetic pathway of rosmarinic acid in Figure 1 based on literature, but their study was not able to explore or confirm a specific enzyme was involved in a specific step of biosynthesis of rosmarinic acid in these two  Salvia species. The enzyme names should be removed from Figure 1.

4) The information from Table 1, 2 and 3 can be found from the contents. So they are redundant. Tables and Figures should present important information such as experiment design and result.

5) Figure 3 can be more-concisely presented by one stacked line with markers plot.

6) In Table 5, why there is no information about other two compounds in leaves? which species was the data from? The author should include complete information or describe the limitation in the figure legend

7) In the conclusion, it is claimed "RAS1 and Cyp98A14 were identified as the rate-limiting genes". Unfortunately, I can't find any evidence to support this conclusion based on the data presented in this manuscript.

Author Response

(The authors gave the same response as above.)

Reviewer 4 Report

In introduction the authors should mark the nutraceutical and antioxidant character of phenols and related references should be added such as:

Durazzo A. Study Approach of Antioxidant Properties in Foods: Update and Considerations. Foods, 02/2017; 6(3):17., DOI:10.3390/foods6030017.

Santini A, Novellino E. Nutraceuticals in hypercholesterolaemia: an overview. Br J Pharmacol. 2017 Jun;174(11):1450-1463. doi: 10.1111/bph.13636. Epub 2016 Oct 29. PMID: 27685833; PMCID: PMC5429323.

Major details should be added in Table 1.

A graphical scheme of sample preparation for qualitative analysis should be added.

Data in Table 3 should be better described.

Data in Figure 3 should be better described and discussed.

Author Response

(The authors gave the same response as above.)

Round 2

Reviewer 3 Report

I think the authors have done a good job to revise their manuscript. The revised manuscript has been greatly improved and qualified to be published.